

# The influence of scopolamine on motor control and attentional processes

Emma Bestaven[1], Charline Kambrun[1], Dominique Guehl[2,3],
Jean-René Cazalets[1] and Etienne Guillaud[1]

[1] Institut de Neurosciences Cognitives et Intégratives d'Aquitaine, UMR 5287 CNRS,
Université de Bordeaux, Bordeaux, France
[2] Institut des Maladies Neurodégénératives, UMR 5293 CNRS, Université de Bordeaux,
Bordeaux, France
[3] Service d'explorations fonctionnelles du système nerveux, Centre Hospitalier Universitaire
de Bordeaux, Bordeaux, France

## ABSTRACT

**Background:** Motion sickness may be caused by a sensory conflict between the visual and the vestibular systems. Scopolamine, known to be the most effective therapy to control the vegetative symptoms of motion sickness, acts on the vestibular nucleus and potentially the vestibulospinal pathway, which may affect balance and motor tasks requiring both attentional process and motor balance. The aim of this study was to explore the effect of scopolamine on motor control and attentional processes.

**Methods:** Seven subjects were evaluated on four different tasks before and after a subcutaneous injection of scopolamine (0.2 mg): a one-minute balance test, a subjective visual vertical test, a pointing task and a galvanic vestibular stimulation with EMG recordings.

**Results:** The results showed that the reaction time and the movement duration were not modified after the injection of scopolamine. However, there was an increase in the center of pressure displacement during the balance test, a decrease in EMG muscle response after galvanic vestibular stimulation and an alteration in the perception of verticality.

**Discussion:** These results confirm that low doses of scopolamine such as those prescribed to avoid motion sickness have no effect on attentional processes, but that it is essential to consider the responsiveness of each subject. However, scopolamine did affect postural control and the perception of verticality. In conclusion, the use of scopolamine to prevent motion sickness must be considered carefully because it could increase imbalances in situations when individuals are already at risk of falling (e.g., sailing, parabolic flight).

Corresponding author
Etienne Guillaud,
etienne.guillaud@u-bordeaux.fr

Parabolic flights, EMG

## INTRODUCTION

Approximately 10–15% of the population experiences motion sickness in many situations, including travelling by road, sea, or air. This trouble can also occur during simulations in a virtual environment, cinema or video games ("pseudo motion sickness").

Indeed, during parabolic flight, motion sickness is also a major issue and the proportion of subjects with severe symptoms is dramatically increased in the absence of adequate pharmacological treatment. A study from *Golding & Denise (2014)* found a far lower rating of motion-sickness in medicated fliers (*Golding & Denise, 2014*) than in non-medicated fliers, and the incidence of vomiting in those medicated was also reduced by half. In all cases, the main symptoms experienced were nausea, vomiting and cold sweat.

Neural mechanisms of motion sickness have been well described in the literature, and the sensory conflict theory (*Reason, 1975*) seems to be a primary explanation for understanding this state. According to this theory, motion sickness results from a discrepancy between the information provided by the visual and the vestibular systems (*Ramos Reis et al., 2013*; *Schmal, 2013*). The motion signals transmitted by the eyes and the vestibular system conflict with each other and with what is expected on the basis of previous interactions with the environment (*Reason, 1978*).

Behavioral as well as pharmacological therapies have been developed to prevent motion sickness. Behavioral strategies, such as avoiding alcohol or coffee intake before travelling and looking at the horizon when sailing, should be prioritized. However, in some extreme conditions when behavioral methods may not be effective, such as during a heavy swell or parabolic flight in an airplane, taking medication may be required. Currently, scopolamine is the most effective drug in controlling the vegetative symptoms of motion sickness. Scopolamine is a natural alkaloid that acts as a non-specific competitive antagonist of cholinergic muscarinic receptors. As such, this drug acts on the parasympathetic system (*Eisenman, 2009*), which explains some of the secondary effects (dry eyes, dry mouth, changes in intraocular pressure, blurred vision, and dizziness (*Brainard & Gresham, 2014*)). In the central nervous system, muscarinic receptors are localized in the cortical and subcortical areas. Besides the dorsal part of the brainstem, the vestibular nuclei (upper, lower and lateral) and the cerebellum were identified as some of the scopolamine binding sites (*Eisenman, 2009*). The action of this drug on the vestibular nuclei suggests that it may also act on motor pathways arising from the vestibular nuclei, in particular the vestibulospinal pathway (*Weerts et al., 2015b*). Since the vestibulospinal pathway regulates body posture and equilibrium (*Uchino & Kushiro, 2011*), people who use scopolamine may present altered capacities for fine tuning of motor balance in normal and challenging conditions. Further, because vestibular processing relies on a network of brain areas whose epicenter is located in the Sylvian fissure and surrounding parieto-temporal and retro-insular regions (*Lopez, Blanke & Mast, 2012*), scopolamine may potentially affect sensorimotor and cognitive processes related to the vestibular system such as self-motion perception, perception of the vertical or visual processing related to gravitational cues.

The aim of the present study was to determine if scopolamine may differentially affect aspects of motor control and/or attentional processes. To answer this question we used several tests: (1) galvanic vestibular stimulation to examine its effect on balance control and lower limb muscle motor responses (*Fitzpatrick & Day, 2004*); (2) analysis of postural stability by means of center of pressure measures (*Duarte & Freitas, 2010*); (3) analysis of subjective visual vertical, which is a test for otolithic output; and (4) a pointing

task that measures reaction time and movement speed. We hypothesized that the scopolamine will disturb balance capacities and will affect the reaction time.

## MATERIALS AND METHODS

### Participants

Seven subjects participated in the experiment (four men and three women, mean age 22.3, SD 2.1). None of them reported vestibular or neuromuscular deficits. The subjects gave their written informed consent and the procedures were in accordance with the ethical standards of the Declaration of Helsinki. The experiments performed in this study were approved by the ethical research committee (Comité de Protection des Personnes Sud Ouest et Outre-Mer III; 2011-A00424-37 27 April 2011).

### Scopolamine administration

Scopolamine (Scopolamine Bromhydrate; Cooper, France) was administered by subcutaneous injections (dosage: 0.2 mg). Subjects were tested before and 30 min after scopolamine injection. All tests were performed under supervision of a medical doctor. In a previous study the maximum serum concentration was found to occur 17.5 min (SD 9.8) after the injection of 0.4 mg of scopolamine (*Ebert et al., 1998*).

### Tests

Subjects passed all tests on the same day. That is why they first carried out the tests in the condition OFF scopolamine and then the tests ON scopolamine. However, the order of the tests in the same condition was randomized to avoid any bias such as fatigue. The mean times between the injection of scopolamine and each test recording are reported in Table 1.

### Pointing task

Subjects were seated in front of a touch screen ($1024 \times 768$) with their finger on a sensor located just in front of their navel. The distance between the sensor and the screen was 42 cm. When the target (a circle) appeared on the screen, they had to touch it as quickly as possible. We tested three different target diameters (10, 20 and 30 mm) with 15 trials for each size. The target position was unpredictable and pseudo-randomized on the entire screen. The reaction time (RT) and the movement duration (MD) were recorded.
All of the processes were performed by a Matlab routine (Matlab R2013a, Mathworks, Natick, USA).

### Subjective visual vertical

The subjective visual vertical (SVV) was assessed using the synapsis SVV device (Synapsys SA, Marseille, France). In a darkroom, a luminous bar was projected on the wall with a video projector. Angular orientation of the bar was changed from trial to trial. The subject was asked to place this bar in a vertical position. The experimenter moved the bar until the subject told him to stop and recorded the angular difference between the actual vertical and the perceived vertical. For each subject, ten trials were recorded

**Table 1 Mean time (*SD*) between scopolamine injection and task performance.**

|  | Mean time (H:MIN) |
|---|---|
| *Pointing task* | 01:05 ± 0:20 |
| *Balance task* | 00:56 ± 0:26 |
| *SVV* | 00:47 ± 0:18 |
| *GVS* | 00:57 ± 0:16 |

and the mean angle value was reported. For each trial, we calculated the deviation from the vertical as the absolute value of the angle measured.

## Posturography

Subjects remained in a quiet erect position for 1 min on a multi-component force platform (AMTI, USA, 50 Hz) to record the position of their center of pressure (CoP). Subjects performed one trial with their eyes open and one trial with their eyes closed. The position of the feet was identical for both tests. Posturographic parameters were selected based on the suggestions of *Prieto et al. (1996)*. The area of the 95% confidence ellipse, the mean velocity of the CoP in the anterior-posterior and in the medial-lateral axis, the standard deviation of the CoP travel distance in the anterior-posterior axis (SD-AP) and in the medial-lateral axis (SD-ML) were calculated with a Matlab routine (MathWorks, Natick, USA).

## Reaction of postural muscles following a vestibular stimulation

To address the effect of scopolamine on balance adjustments that follow vestibular perturbations, we recorded electromyography (EMG) on bilateral Erector Spinae and Gastrocnemius Medialis during Galvanic Vestibular Stimulation (GVS). To apply GVS, two electrodes were positioned on the mastoid processes (Pals platinum, round, 8 cm$^2$). In the binaural configuration (see review by *Fitzpatrick & Day (2004)*), passing an electric current of 3 mA between the electrodes increases the activity of the vestibular part of the vestibulocochlear nerve (VIII) located at the cathode side and decreases its activity on the anode side (*Goldberg, Smith & Fernandez, 1984*). This artificial change in vestibular nerves is similar to that produced by sway toward the cathode side (*Cathers, Day & Fitzpatrick, 2005*). In our experiment, subjects were asked to turn their head to the right. Using a bipolar constant current generator (Digitimer DS5, Letchworth Garden, United Kingdom), binaural GVS was applied with the cathode on the left side. In this configuration, the subjects experienced an illusion of forward sway (slightly left forward because the head was not strictly turned 90° to the right). The expected response is backside muscles activation (trunk and lower limbs extensors) with latencies between 50–200 ms (*Britton et al., 1993*; *Cathers, Day & Fitzpatrick, 2005*; *Fitzpatrick, Burke & Gandevia, 1994*).

EMG from bilateral medial Gastrocnemius and Erector spinae was recorded at 1,000 Hz during the head-turned balance task using an analogical amplifier (TeleEMG, BTS, Milano, Italy) linked to an ITC-18 A/D card (Heka, Lambrecht, Germany). The same A/D

card was used to start the GVS stimulation and record the EMG signals; all of the processes were performed by Matlab R2013a (Mathworks, Natick, MA, USA).

Each subject (n = 7) received 60 GVS (bipolar binaural square pulse, 3 mA, 170 ms) per condition (OFF scopolamine, ON scopolamine) spread over 15 sets. Each set consisted of 4 GVS that occurred with randomized delay (inter-stimulation delays were greater than 5 s). Between the trials sets, subjects could move their head and have a break if desired. Because visual cues are known to reduce the GVS response amplitude (*Cathers, Day & Fitzpatrick, 2005*; *Lund & Broberg, 1983*), subjects were blindfolded.

EMG signals were analogically amplified (x1k), numerically high-pass filtered (reverse Butterworth filter at 30 Hz), and rectified. For EMG signals from Erector Spinae, GVS artifacts (shifts due to voltage) were removed a posteriori by digitalized data shift compensation. For each subject and each stimulation, a 4 s range was defined around the GVS start (2 s before and 2 s after). All of the EMG ranges were synchronized and averaged to have one sample of 4 s per subject, per muscle and per scopolamine condition. These samples were filtered by low-pass filtering (reverse Butterworth filter at 10 Hz), and normalized by the activity measured at the trials beginning (tonic activity of normal standing = 100%). The EMG medium latency response to the GVS (ML; *Britton et al., 1993*) were visually identified. The start and end delays were measured, ML EMG area and peak amplitude were computed and submitted to statistics.

## Statistics

The results are presented as the mean and standard deviation (SD). For the posturographic tasks, two-way repeated measures ANOVA was used to test the effect of the condition (Open Eyes vs. Closed Eyes) and of scopolamine (OFF vs. ON). For the pointing task, two-way repeated measures ANOVA was used to test the influence of the target size (10, 20 and 30 mm) and of scopolamine (OFF vs. ON). For the GVS, three-way repeated measures ANOVA was used to test the influence of muscle (Erector Spinae vs. Gastrocnemieus Medialis), scopolamine (OFF vs. ON), and laterality (left muscle vs. right muscle). Finally for the SVV, paired T-test were used to analyze possible differences between the OFF and ON conditions. Correlations were performed using Spearman's correlation coefficient (Rho).

Results were considered statistically significant for $P < 0.05$. Statistical analysis was performed with IBM SPSS Statistics Version 20 (IBM Corporation, USA).

## RESULTS

### Pointing task

Figure 1 illustrates the results for the RT and the MD. To assess if there was an effect of practice we correlated the trial number with the variables RT and MD, but did not find, however, any relationship. For the reaction time, the Spearman's correlation coefficient was Rho = −0.07 ($P = 0.2$) for the OFF condition and Rho = 0.036 ($P = 0.5$) for the ON scopolamine condition. Similar findings were obtained for the movement duration in OFF and ON scopolamine condition (respectively Rho = −0.032 ($P = 0.6$) and Rho = −0.072 ($P = 0.2$)).

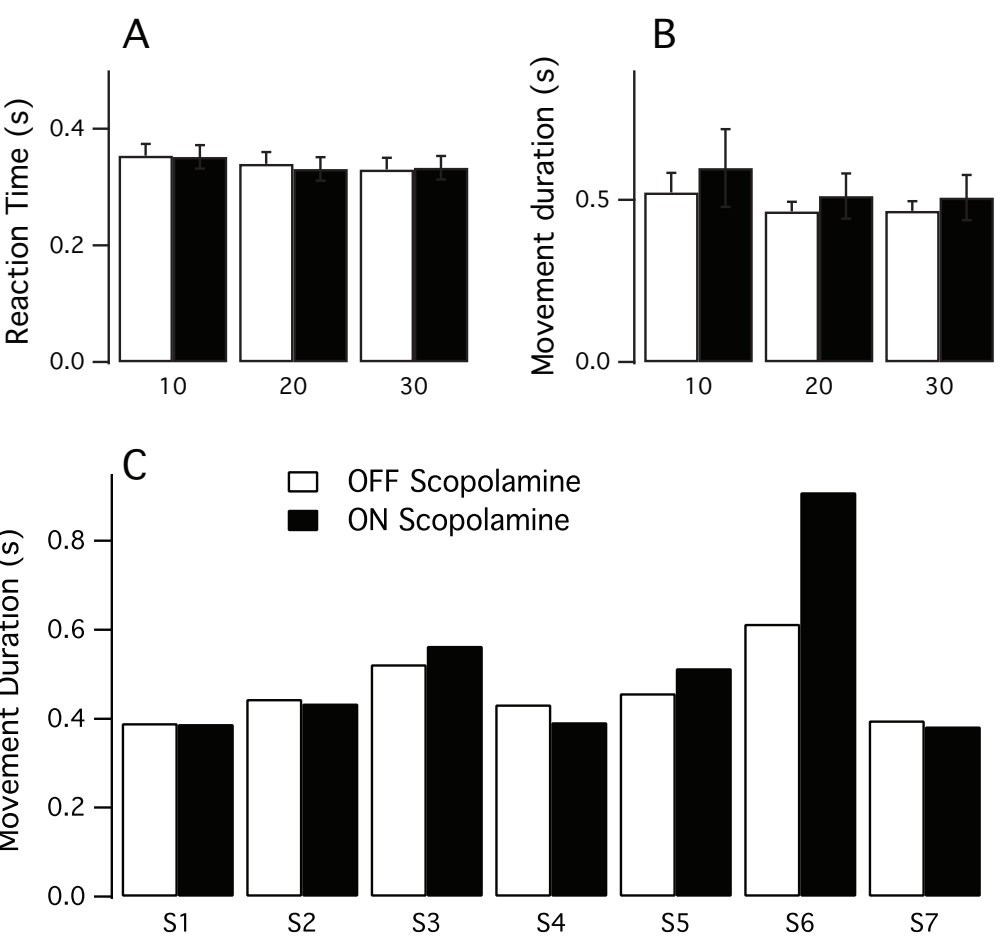

**Figure 1 Results of the pointing task.** (A) Mean reaction time in seconds for each target size (10, 20 and 30 mm) in both conditions (ON and OFF scopolamine). (B) Mean movement duration in seconds for each target size (10, 20 and 30 mm) in both conditions ON and OFF scopolamine. (C) Movement duration in seconds ON and OFF scopolamine for each subject (n = 7, target size = 20 mm). Subject 6 (S6) greatly increased movement duration after the injection of scopolamine.

Multivariate analysis showed no effect of scopolamine on the RT (Mean_OFF 0.341 s, SD 0.01; Mean_ON 0.339 s, SD 0.02; $F(1,6) = 1.048$, $P = 0.345$, Fig. 1A) or on the movement duration (Mean_OFF 0.485 s, SD 0.03; Mean_ON 0.540 s, SD 0.05; $F(1,6) = 1.171$, $P = 0.321$, Fig. 1B). However, there was a significant effect of the target size; the reaction time was longer for smaller targets (Mean_10 0.353 s, SD 0.05; Mean_20 0.336 s, SD 0.05; Mean_30 0.331 s, SD 0.04; $F(2,5) = 13.971$, $P = 0.009$).

Individual data revealed some disparities, particularly in movement duration. As illustrated in Fig. 1C, although subject 6 had the expected results in the OFF scopolamine condition, the movement duration results increased substantially (MD_OFF 0.458 s; MD_ON 0.514 s) after the injection of the drug.

## Subjective visual vertical

All subjects had normal results during the OFF scopolamine condition (range from −1.74–1.77°) with the pathological limit being fixed to 3° by Synapsys. During the ON

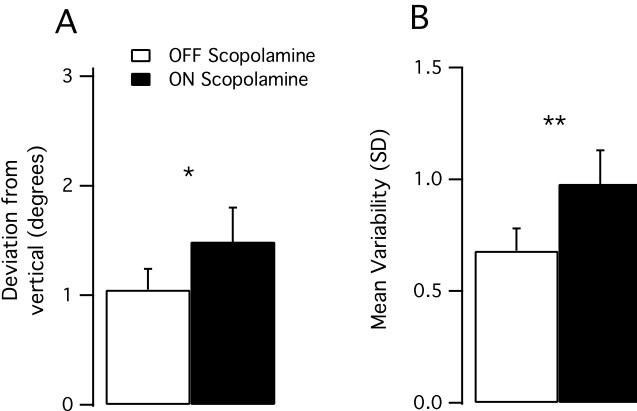

**Figure 2 Results of the subjective visual vertical test.** (A) Mean deviation from the real vertical in degrees. The deviation significantly increased after the injection of scopolamine ($P < 0.05$). (B) Mean trial-to-trial variability (standard deviation) increased in the ON scopolamine condition.

scopolamine condition, values ranged from $-2.33°–3.18°$. The mean deviation to the real vertical (absolute value) was higher in the ON scopolamine condition than in the OFF scopolamine condition (Mean_OFF $1.06°$, SD 0.6, Mean_ON $1.59°$, SD 0.9, $P = 0.047$, Fig. 2A).

We tested the trial-to-trial variability (10 trials per subject) and found that the subjects' responses were more scattered after the injection of scopolamine, as the SD of the performance increased (Var_OFF 0.674, Var_ON 0.992; $P = 0.01$; Fig. 2B).

## Postural control (Fig. 3)

We computed the posturographic parameters during 1-min balance tests with eyes opened then with eyes closed. We observed an effect of the scopolamine on the area of the stabilogram ($F(1,6) = 7.171$; $P = 0.037$). With the eyes open, the mean area was 88.8 mm$^2$, SD 47.6 in the OFF scopolamine condition, and was 127.4 mm$^2$, SD 97.5 in the ON scopolamine condition (Fig. 3A). With the eyes closed, the mean area was 100.4 mm$^2$, SD 57.9 in the OFF scopolamine condition, and was 135.2 mm$^2$, SD 43.7 in the ON scopolamine condition (Fig. 3A).

The SD-AP was also significantly larger following the injection of scopolamine ($F(1,6) = 57.6$, $P < 0.001$; Mean_SD-AP_OFF_EyesClosed 4.9 cm, SD 1.5; Mean_SD-AP_ON_EyesClosed 5.9 cm, SD 0.7; Figs. 3B and 3C). No effect of the scopolamine was observed on the SD_ML or on the mean velocity of the CoP.

## Galvanic vestibular stimulation

The most important result from the EMG response to GVS was the effect of scopolamine on the response area ($F(1,6) = 10.39$, $P < 0.05$; Fig. 4A). From OFF scopolamine to ON scopolamine condition, the ML response area decrease was 32.2% for the Gastrocnemius and 44% for the erector spinae (in arbitrary units, Gastroc_off = 62.2, Gastroc_on = 20; ES_off = 18.8; ES_on = 10.8; Fig. 4B). The "muscle" main effect was also significant (Gastrocnemius vs. Erector Spinae, $F(1,6) = 10.49$, $P < 0.05$), with a larger area for the

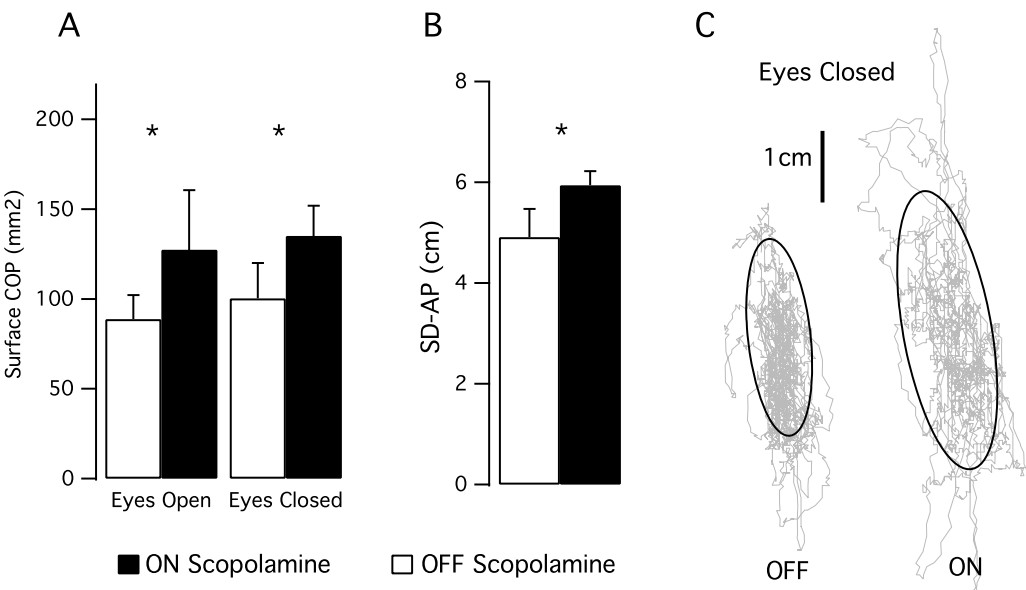

**Figure 3 Results of the posturographic tests.** (A) Mean surface of the stabilogram during a 1 min balance test with eyes opened then with eyes closed. Effect of the scopolamine was significant ($P = 0.037$). (B) The histogram shows the mean standard deviation in the anterior-posterior axis (SD-AP) of the CoP displacement during a 1 min balance test with eyes closed. (C) Stabilograms for a representative subject OFF scopolamine and ON scopolamine during a 1-min balance test with eyes closed. The grey curves represent the CoP displacement, and the black curves are the 95% confidence ellipse.

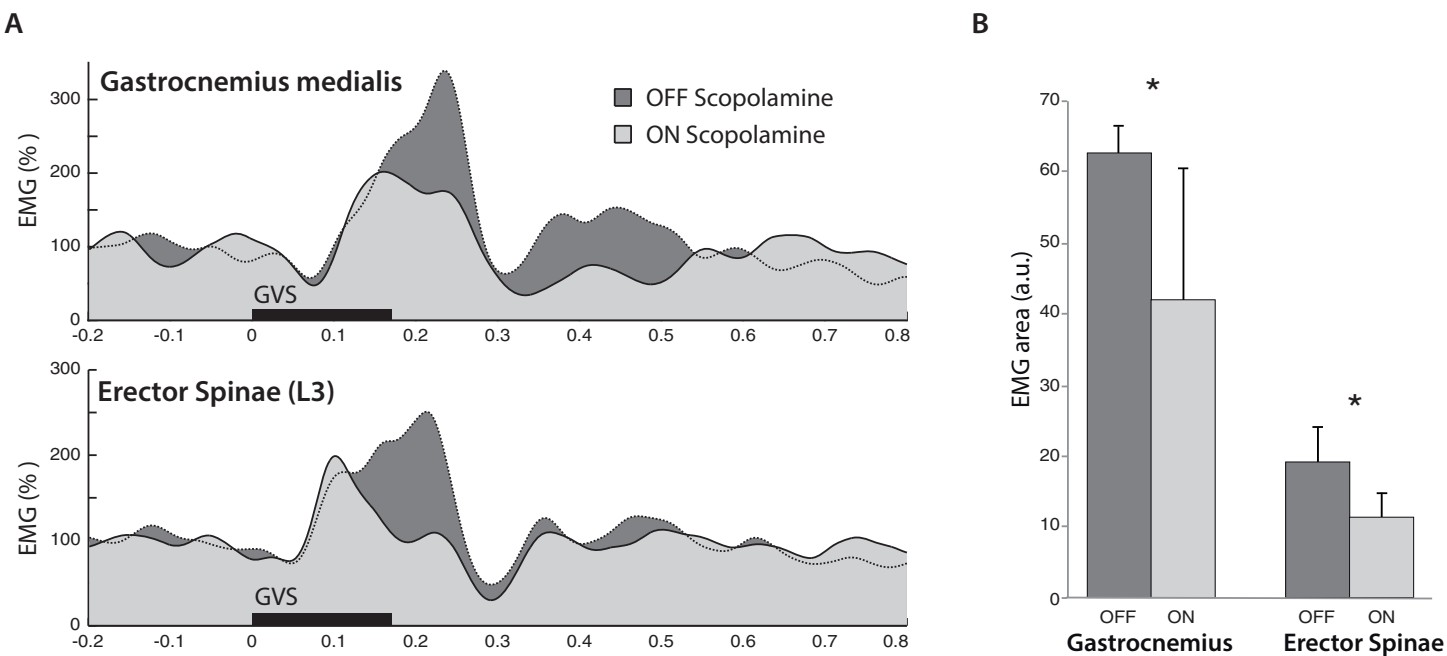

**Figure 4 Electromyography following GVS.** (A) Averaged EMG signals from one typical subject in both Gastrocnemius medialis and Erector Spinae. Trials were synchronized on GVS start ($t = 0$), and the muscle activity increased at medium latencies (from 50–300 ms). In both muscles, scopolamine administration reduced the amplitude of the response. (B) Mean EMG response area at medium latency from all subjects. In both muscles, the ML response was reduced in ON scopolamine condition.

Gastrocnemius than the ES (52 vs. 14.8 a.u., respectively). Similar effects were observed for the maximal amplitude of the ML response.

The primary effect of the muscle was observed in the time to peak ($F_{(1,6)} = 7.97$, $P < 0.05$), with a maximal EMG reached more quickly in the Erector Spinae than in the Gastrocnemius (130 vs. 196 ms after GVS start, respectively). In addition to the observed effect on time to peak, we observed a similar (but not significant, $F_{(1,6)} = 1.86$, $P = 0.22$) tendency on the start time of the ML response with delays of 79 ms for the Gastrocnemius vs. 50 ms for the ES. A shorter delay for the ES than for the Gastrocnemius in ML response initiation has been reported by *Ali, Rowen & Iles (2003)*, with latencies of 60 vs. 85 ms.

## DISCUSSION

In the present study, we assessed the effects of scopolamine using a before-after approach on various tasks requiring visuo-motor coordination processes, attention, and balance. Although the lack of placebo control group is a limitation of the study design, we found that 0.2 mg of scopolamine did not affect reaction time and movement duration in a pointing task, while in the tasks that challenged the vestibular system (subjective visual vertical, posturography, vestibulospinal reflex elicited by galvanic stimulation), it elicited significant changes.

Several studies (*Ellis et al., 2006*; *Ramos Reis et al., 2013*; *Simmons et al., 2010*; *Weerts et al., 2015b*; *Wesnes & Warburton, 1984*) have evaluated the effects of various doses of scopolamine on cognitive tasks, and partly contradictory results have been reported, likely related to methodological differences between studies. Despite the low number of subjects which could prevent from detecting subtle changes, we found that a 0.2 mg dose of scopolamine did not affect the reaction time and the movement duration (Figs. 1A and 1B). We used this dose because scopolamine is usually prescribed between 0.05–0.2 mg to prevent motion sickness (*Golding & Denise, 2014*). Our results are consistent with those of *Weerts et al. (2015b)*, who did not observe changes in RT in a psychomotor vigilance task with a simple RT (click on a mouse button at visual signal). Similarly, *Simmons et al. (2010)* did not find significant effects on RT in a substitution/delayed recall task. In these two experiments, the subjects inhaled 0.4 mg of scopolamine by intranasal spray. Although a direct comparison between intranasal and subcutaneous scopolamine administration (as in the present study) should be made cautiously, it is likely that the 0.4 mg dose of inhaled scopolamine would be closer to the dose used in this study because subcutaneous delivery results in significantly higher bioavailability and higher rate of absorption. The use of higher doses of scopolamine, however, has been found to have an effect on rapid information processing (*Wesnes & Warburton, 1984*). In a task of consecutive symbol detection, RT was affected by scopolamine at a dose of 1.2 mg, whereas a lower dose of 0.6 mg did not result in changes. Noticeably, in our experiment, although we injected the same dose for all of the subjects, a female subject presented an exacerbated response. She not only reported that she felt very drowsy, but there was a major effect of scopolamine on her movement time (see subject 6 on Fig. 1C). Altogether, this indicates that for highly responsive subjects and/or at higher doses,

one might expect a decrease in attention or cognitive process, but with the usually prescribed posology for motion sickness prevention (0.05–0.2 mg), the vigilance tasks will not be significantly affected.

To our knowledge, our study is the first to explore the impact of the scopolamine on human balance performance. When the subjects were requested to remain standing in a stationary position, both the CoP area and the anterior-posterior oscillation amplitude were significantly increased after scopolamine administration (Fig. 3). According to these results, scopolamine prescribers should be vigilant to the elderly about the risks of falls which are already increased by age-related changes in the sensorimotor and neuromuscular systems.

However, we found a higher inter-subject variability in the eyes-open condition (see error bar Fig. 3A, SD = 97.5 ON scopolamine). This higher variability in eyes-open condition could be explained by the perceptual styles of the subjects, with a stronger effect of scopolamine on subjects who use somato-sensory cues preferentially (*Isableu & Vuillerme, 2006*; *Kluzik, Horak & Peterka, 2005*) than on ones with visual preference (*Chiari, Bertani & Cappello, 2000*; *Isableu et al., 1997*). In the eyes closed condition, in which only somato-sensory (including vestibular) cues were available, the perceptual style would have resulted in lower inter-subject variability. This would be in accordance with a recent study (*Weerts et al., 2015b*) that provided a direct functional demonstration of the impact of scopolamine on cues that originated from the vestibular semi-circular canals and the utricles. These authors observed a reduction in the gain of the vestibulo-ocular reflex and a decrease in the total caloric response after scopolamine administration (0.4 mg intranasally). These results could be explained by the binding sites for scopolamine on the vestibular nucleus (*Jaju, Kirsten & Wang, 1970*; *Jaju & Wang, 1971*; *Matsuoka, Domino & Morimoto, 1975*; *Pyykko et al., 1984*), potentially in the inner ear end organs (vestibular hair cells) and in the vestibular nerve fibers (*Li, Chun & Ju, 2007*; *Weerts et al., 2015a*). Moreover, some authors (*Kushiro et al., 2008*; *Uchino & Kushiro, 2011*) have suggested that the vestibular projections that reach the lower part of the spinal cord (L3) mostly originate from the posterior semi-circular canal. The vestibular inputs are conveyed by the lateral part of the vestibular nucleus (*Matsuoka, Domino & Morimoto, 1975*), and this pathway could be affected by scopolamine from the hair cells to the spinal cord (*Woolf, 1991*).

The vestibular vagueness induced by scopolamine also appears to have affected the subjects' precision in the subjective visual vertical task (Fig. 3). After scopolamine injection, the mean deviation between SVV and gravitational vector was increased, as well as the subjects' trial-to-trial variability. As SVV provides an indirect assessment of the otolithic function, this also suggests a greater difficulty in precisely accounting for the sensory cues required to estimate the vertical. Various studies that tested the role of the vestibular system in orientation in patients with vestibular impairments (*Borel et al., 2008*) have shown that there was a perceived subjective visual vertical tilted toward the side of the lesion (*Friedmann, 1970*; *Vibert & Hausler, 2000*). Thus, with regard to our postural tasks, this lack of precision after scopolamine injection could be likely attributed to an altered vestibular input.

The GVS test used in the present study directly challenged the balance control originating from vestibular inputs only (*Britton et al., 1993*; *Fitzpatrick & Day, 2004*). Following scopolamine administration, we observed a strong reduction of extensor muscle contraction in response to GVS, which suggests a scopolamine action on vestibulospinal processes. Together with the reported effect on CoP oscillations (Fig. 2), these results indicate that scopolamine administration may alter functional body balance capacities, even with the low doses (0.2 mg) used for the motion sickness. Although it is difficult to estimate the induced risk for astronauts after scopolamine absorption (to prevent spatial motion sickness), the balance alteration could have more dramatic consequences for people submitted subjected to gravity under terrestrial conditions, such as sailors or parabolic-flight fliers. Similarly, as the vestibulospinal pathways support a large range of motor behavior, such as arm movement correction during unpredicted body displacement (*Blouin et al., 2015*; *Bresciani et al., 2002*; *Guillaud, Simoneau & Blouin, 2011*), scopolamine could affect the control of reaching and grasping movements.

## CONCLUSIONS

In conclusion, the use of scopolamine to prevent motion sickness and spatial motion sickness must be considered carefully. Despite the fact that scopolamine has been described as the most effective for motion sickness, our study presents an effect of scopolamine on the vestibulospinal pathway, even at lower doses. Scopolamine could be the source of imbalance in gravitational environments and clumsy motor acts. These effects could be explained by its action on cholinergic muscarinic receptors in the vestibular system.

### Funding
This work was supported by the Centre National d'Etudes Spatiales (CNES). It is based on ground observations parallels to the VP92, VP95 and VP102 campaign. The funders had no role in study design, data collection and analysis, decision to publish, or preparation of the manuscript.

### Competing Interests
The authors declare that they have no competing interests.

### Author Contributions
- Emma Bestaven conceived and designed the experiments, performed the experiments, analyzed the data, contributed reagents/materials/analysis tools, wrote the paper, prepared figures and/or tables, reviewed drafts of the paper.
- Charline Kambrun conceived and designed the experiments, performed the experiments, analyzed the data, contributed reagents/materials/analysis tools, wrote the paper, prepared figures and/or tables, reviewed drafts of the paper.

- Dominique Guehl conceived and designed the experiments, performed the experiments, analyzed the data, contributed reagents/materials/analysis tools, wrote the paper, prepared figures and/or tables, reviewed drafts of the paper.
- Jean-René Cazalets conceived and designed the experiments, performed the experiments, analyzed the data, contributed reagents/materials/analysis tools, wrote the paper, prepared figures and/or tables, reviewed drafts of the paper.
- Etienne Guillaud conceived and designed the experiments, performed the experiments, analyzed the data, contributed reagents/materials/analysis tools, wrote the paper, prepared figures and/or tables, reviewed drafts of the paper.

### Human Ethics

The following information was supplied relating to ethical approvals (i.e., approving body and any reference numbers):

Comité de Protection des Personnes Sud Ouest et Outre-Mer III; Approval Number: 2011-A00424-37 27 April 2011.

### Data Deposition

The raw data is available at:

https://bfs.u-bordeaux.fr/telecharge.php?choix=files/00ddbe73120f8ac46cb16d2184b683d9/RawData.zip.

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
