# Peer review of "The influence of scopolamine on motor control and attentional processes"

_PeerJ, doi:10.7717/peerj.2008_

## Round 0.1 · original submission · Major Revisions

· Academic Editor

Major Revisions

Since we were only able to find one reviewer, I reviewed the manuscript myself based on my own expertise. Like the other reviewer, I think that the paper is well written and is the first study to address the effects of a low dose of scopolamine on motor control and attention. I see however a major drawback in the design that was not placebo controlled and the low number of subjects. Hence, the conclusion that a low dose of scopolamine has no effect on attentional processes cannot be made from the data. First, since there was no placebo control the effects of scopolamine were tested in a before-after approach. The absence of effects on RT of scopolamine may be due to an interaction of practice which would decrease RT and an increase in RT by scopolamine. Second, most tasks administering scopolamine subcutaneously used a 60 min wait period prior to testing since maximal effects on alertness and attention or other central measures occur between 1-2 hrs post injection (e.g. Ebert et al. Clin Pharmacol. 1998). The 30 min interval may have been too short to observe effects on RTs. Third, the effects on attention may be smaller in size than effects on posture, hence 7 subjects may be just to low to detect a significant change. Hence, these confounds need to be addressed in any revised version.

The effects on balance are also highly relevant to older individuals taking scopolamine, this should be discussed.
l.74 there are more scopolamine binding sites in the central nervous system.
l.81 in my opinion, the effect on cognitive processes is due to the effect of scopolamine on the projection from the basal forebrain to the cortex, rather than vestibular processes being located in parieto-temporal regions.
For reporting ANOIVA results F values and degrees of freedom should be consistently reported. Which test do the stars on figures 2 and 3 reflect. Where these post hoc t-test?

Reviewer 1 ·

Basic reporting

This is a well-written paper about the effects of scopolamine on attention and motor control. There is good justification for doing the study and a relevant literature review is presented to contextualize the paper.

Experimental design

The methods are clear and appear to be replicable by another investigator. The cognitive and motor tasks as well as the medication administration are clearly described.

Validity of the findings

The statistics are appropriate given the study design. The conclusions are appropriate given the results.

Additional comments

This is a well-written paper that makes an interesting contribution to the effects of scopolamine on cognitive and motor functioning. As the authors note it is the first study of scopolamine effects on motor control and this is an important finding.

---

## Round 0.2 · Minor Revisions

· Academic Editor

Minor Revisions

I see the effort to show that there are no session effects, however this cannot be calculated from the present data. First it is invalid to do a Person correlation using the test session variable which is data on the nominal scale. Second, session is always confounded by drug. So either the authors have to test a couple of new subjects without drug twice or state this as well as the lack of a placebo control as a limitation.

With respect to scopolamine binding you only mention the brain stem and cerebellum, I think cortical and subcortical receptors should at least be acknowledged.

---

## Round 0.3 · accepted · Accept

· Academic Editor

Accept

All comments were adressed.